# Evolution of Sweet Potato (*Ipomoea batatas* [L.] Lam.) Breeding in Cuba

**DOI:** 10.3390/plants14131911

**Published:** 2025-06-21

**Authors:** Alfredo Morales, Peiyong Ma, Zhaodong Jia, Dania Rodríguez, Iván Javier Pastrana Vargas, Vaniert Ventura, José Efraín González, Orelvis Portal, Federico Diaz, Oscar Parrado Alvarez, Carina Cordero, Xiaofeng Bian

**Affiliations:** 1Institute of Food Crops, Jiangsu Academy of Agricultural Sciences (JAAS), Nanjing 210095, China; alfremr88@gmail.com (A.M.); andympy@163.com (P.M.); jzdgood162@126.com (Z.J.); 2Research Institute of Tropical Roots and Tuber Crops (INIVIT), Santo Domingo 53000, Cuba; daniarodriguezdelsol@gmail.com (D.R.); vaniertvc1983@gmail.com (V.V.); gefrain00@gmail.com (J.E.G.); 3Department of Agronomic Engineering and Rural Development, Faculty of Agricultural Sciences, Universidad de Córdoba, Carrera 6 No. 77-305, Montería 230002, Colombia; ivanpastranav@correo.unicordoba.edu.co; 4Departamento de Biología, Facultad de Ciencias Agropecuarias, Universidad Central “Marta Abreu” de Las Villas, Santa Clara 54830, Cuba; orelvispv@uclv.cu; 5Centro de Investigaciones Agropecuarias, Facultad de Ciencias Agropecuarias, Universidad Central “Marta Abreu” de Las Villas, Santa Clara 54830, Cuba; 6International Potato Center (CIP), Lima 15023, Peru; federdt76@gmail.com; 7Departamento de Educación Agropecuaria, Facultad de Ciencias Agropecuarias, Universidad de Camagüey Ignacio Agramonte Loynaz, 94Q9+H5G, Carretera Central, Camagüey 74650, Cuba; agrisost2017@gmail.com; 8Corporación Colombiana de Investigación Agropecuaria—AGROSAVIA, Centro de Investigación Motilonia, Km 5 vía a Becerril, Agustín Codazzi 202050, Colombia; ccordero@agrosavia.co

**Keywords:** sweet potato breeding, genetic gain, yield improvement, stress tolerance, phenotypic selection

## Abstract

This study analyzed the genetic progress of sweet potato (*Ipomoea batatas*) breeding in Cuba over the past 50 years by field trials comparing traditional and improved varieties. Improved varieties significantly outperformed traditional ones in tuberous root yield, with an accumulated genetic gain of 0.20–0.37 t ha^−1^ per year, translating to a 256% yield increase. Improved genotypes also exhibited enhanced pest tolerance: lower weevil (*Cylas formicarius*) infestation and reduced nematode (*Meloidogyne incognita*) reproduction rates. For viral diseases, 60% of improved varieties showed incidence rates below 10%, compared with 90% of traditional varieties exceeding this threshold. Under drought conditions, improved varieties showed tolerance, with Stress Susceptibility Indices (SSIs) of less than 0.8, while the traditional varieties were more susceptible (SSI > 1). Phenotypic stability analysis via GGE biplot confirmed the superior yield and adaptability of improved varieties across environments. These advances underscore the critical role of sweet potatoes breeding in Cuba, with improvements in yield, quality and resistance to biotic and abiotic stress, contributing to strengthening climate resilience and food security.

## 1. Introduction

Sweet potato (*Ipomoea batatas* [L.] Lam.) ranks among the oldest domesticated crops, with radiocarbon evidence tracing its origin to ~8080 BC (± 170 years) in the Tres Ventanas caves of the Chilca Canyon in South-central Peru [1]. This was a process of empirical plant breeding carried out over millennia, in which early farmers relied on the genetic variation present in wild plant populations and selected individual plants with the desired traits. This resulted in the agricultural species grown today [2].

The origin of modern plant breeding dates back to 1865, when Gregor Mendel described how specific trait factors are related in peas (*Pisum sativum* L.). His experiments led to the formulation of the basic rules of inheritance. His work was rediscovered in 1900 and confirmed by E. von Tschermak, C. Correns and H. de Vries, thereby laying the foundations of modern genetics, and researchers began to cross superior varieties and create hybrids with the desired characteristics [3,4].

Modern genetic improvement (post-Mendelian) has led to significant yield increases in crops such as maize, rice, sorghum, wheat and soybean. In the United States, for example, maize yields increased from about 2 t ha^−1^ in the 1940s to 7 t ha^−1^ in the 1990s. In England, it took only 40 years for wheat yields to increase from 2 to 6 t ha^−1^ [5,6]. Furthermore, in 1965, Norman E. Borlaug pioneered the “Green Revolution” by using plant genetics to generate semi-dwarf, fertilizer-sensitive, disease-resistant plants [7]. However, while the “Green Revolution” transformed cereals, tropical root crops like sweet potato have lagged behind in genetic innovation, partly due to their complex genetics, including hexaploidy and polysomic inheritance [8].

Early efforts to address this gap included Julian Miller’s research work in 1937 at Louisiana State University in the United States on true botanical seed and growing seedlings, regarded as one of the first successful sweet potato breeding programs in the world [9].

Parallel developments occurred in Cuba, where early records by Alvaro Reynoso (1867) documented sweet potato diversity, and the work of Juan Tomas Roig in the early 20th pioneered the identification of high-yielding genotypes (>30 t ha^−1^) [10]. After this period, despite the scarce research developed due to limited funding, it was possible to select high-yield genotypes such as ‘Blanco Redondo’ by the Centro de Investigaciones Agropecuarias (CIAP) of the Universidad Central “Marta Abreu” de Las Villas (1962–1966), and ‘Cuba 1’, ‘Cuba 2’, ‘Cuba 3’, ‘Cuba 6’, ‘Cuba 9’, ‘Haiti’ and ‘Baracutey’ (1967–1972) by the Centro de Mejoramiento de Semilla Agámica (CEMSA) “Fructuoso Rodríguez” (Today the Research Institute of Tropical Roots and Tuber Crops—INIVIT) [11].

It was not until 1972 that Dr. Alfredo Morales Tejón, at INIVIT, created the first successful Genetic Improvement Program (GIP) in Cuba through directed crossings (hybridizations), which had a defined purpose with specific objectives. The first requirement of this program was to collect all the possible genetic variations present in Cuba and in different countries of the world. Thus, in the 1980s, the Cuban National Collection of Sweet Potato Germplasm was created, with 731 accessions: 387 natives, 146 foreign (China, Japan, Vietnam, United States, different Caribbean islands, Panama, Nicaragua, Colombia, Peru, Brazil, Spain and Nigeria) and 198 improved. This collection became the most important in Central America and the Caribbean due to the number of materials conserved and the wide genetic diversity it possesses.

Once this genetic variation was obtained, the elite parents and the best combinations between them were identified and selected. For more than 50 years, sweet potato breeding in Cuba has been based on conventional breeding, which is a lengthy process that generally takes around 5–6 years to release a new cultivar [8]. Furthermore, regardless of the final objective for the improved cultivar, whether for human or animal consumption or for industry, the cultivars released by INIVIT must always have certain general characteristics that are necessary, such as high yield, good agronomic characteristics (high number of tuberous roots, good take-up, etc.), stability and adaptability, tolerance to biotic and abiotic stress factors, organoleptic quality and early maturity.

This conventional breeding (based solely on mass phenotypic selection without the use of tissue culture or molecular markers) relies exclusively on the phenotypic expressions of the genotypes to identify the superior ones. For example, most of the genotypes had defects on the surface of the tubers, were unstable or very susceptible to pests and diseases; moreover, it was very difficult to combine all the desired characteristics in the same genotype. This is because the sweet potato is a hexaploid species (2n = 6x = 90), with the basic number x = 15 [12]. It follows the polysomic inheritance route, with possible disomic, tetrasomic, tetradisomic and hexasomic models [13]. Furthermore, at 6x within the allele frequency range of about q = 0.2 to q = 0.8, the heterozygosity frequency is still >0.75 [14], so the genotypes used as parents are highly heterozygous. It also has a large and complex genome of about 2–3 Gb in size [15]. There is also a large number of possible genotypes in segregating populations due to the various combinations of parental chromosomes provided by the male and female gametes [16]. As calculated by Yan et al. [17], the allelic combinations in F1 progeny exceed 10^39^, reflecting the extreme heterozygosity of hexaploid sweet potato.

The conventional breeding scheme used by INIVIT has consisted of open pollination designs and directed crossing blocks, using 30 to 40 parents with as many superior traits as possible, obtaining large hybrid segregating populations annually (more than 40,000), using a high-precision scheme (based on phenotypes) to identify and select transgressive segregants with desired trait combinations. The GIP, led by Dr. Morales (1972–2019), has released 24 commercial cultivars of sweet potato, covering 98% of the area dedicated to sweet potato production in Cuba. The most notable cultivar was the obtainment of genotype number I01-B2, which was released in 2005 under the commercial name ‘INIVIT B2-2005’. This cultivar produces an average of 7 t ha^−1^ more than any other commercial cultivar grown in Cuba. Its adoption by farmers has contributed greatly to an increase in the national average yield from 4 to 11 t ha^−1^. ‘INIVIT B2-2005’ occupies 52% of the total sweet potato area in Cuba. In addition, several improvements have been achieved, such as attractive skin and flesh colors, higher carotene and anthocyanin content, greater stability and adaptability and tolerance to some pests (*Cylas formicarius* Fab., *Typophorus nigritus* F. and nematodes) [11]. The above shows that the INIVIT program has overcame the challenges of hexaploidy without molecular tools, only by massive phenotypic selection.

While significant progress has been made in breeding diploid crops, quantitative assessments of tropical polyploid improvement under technology-limited conditions remain scarce. This study presents the first comprehensive quantitative analysis of a 50-year sweet potato breeding program conducted under tropical conditions without molecular tools, providing a replicable model for resource-constrained regions. Specifically, we address the critical question: What level of genetic gain in productivity and stress tolerance has been achieved through sweet potato breeding in Cuba?

## 2. Results

### 2.1. Progress in Agronomic Traits

No significant interaction was observed between the factors (Season x Cultivars) for the response variables (*p*-value < 0.05). Significant differences were detected between seasons (Spring and Winter) for yield, commercial yield, foliage yield and dry matter (*p*-value < 0.05), while no significant differences were found for the number of tuberous roots and °Brix.

For the cultivars factor (traditional and improved), the results of the analysis of variance provide statistical evidence to conclude that there were no significant differences for the response variables dry matter and °Brix, with a reliability of 95% (*p*-value > 0.05) (Table 1).

The yield of tuberous roots was influenced by the planting season, with statistically significant differences between the two seasons (*p*-value < 0.05). The sweet potato plantation planted in winter had a yield of 21.59 t ha^−1^, which was 6.87 t ha^−1^ higher than the spring yield, where cultivars yielded 46.67% more in winter. The improved sweet potato genotypes had a higher yield than the traditional genotypes in both winter and spring, with statistically significant differences at a significance level of 0.05. In winter, the yield of the improved genotypes was 121% (16.31 t ha^−1^ more) higher than the traditional genotypes, while in spring it was 311% (17.92 t ha^−1^) higher. Based on the above, the average yield of the improved genotypes between the two seasons is 17.11 t ha^−1^ (178%) higher than the traditional genotypes (Figure 1).

Commercial tuberous root yield by planting season was higher in winter, with statistically significant differences (*p*-value < 0.05). The yield in winter was 14.85 t ha^−1^, 5.62 t ha^−1^ higher than the spring yields. When comparing the groups of genotypes, the improved genotypes had a commercial yield 14.32 t ha^−1^ higher than the traditional genotypes (Figure 2).

There were no statistically significant differences between seasons (*p*-value < 0.05) for the number of tuberous roots trait. However, there were statistical differences between genotype groups at a significance level of 0.05. The improved genotypes averaged 3.69 tuberous roots per plant, and the traditional genotypes averaged 1.93 (Figure 3).

Foliage yield was 4.45 t ha^−1^ higher in spring than in winter, with statistically significant differences. Traditional genotypes had a higher foliage yield of 2.48 t ha^−1^ than improved genotypes, with statistically significant differences (Figure 4).

Dry matter in tuberous roots was higher in winter, with an average dry matter content of 27.40%, 1.84% higher than in spring. However, there were no statistically significant differences between the genotype groups (Figure 5).

The total soluble solids content of tuberous roots was not statistically influenced by either planting season or genotype group. However, there was a slight difference of 0.61 °Brix more in winter than in spring. Furthermore, the traditional genotypes averaged 9.85 °Brix, which is 0.43 °Brix higher than the improved genotypes (Figure 6).

Storage roots initiation was influenced by genotype group, with statistically significant differences (*p*-value < 0.05). Improved genotypes initiated storage roots at an average of 33.06 days, with a range of 25–42 days within this group. Traditional genotypes initiated storage roots at an average of 42.16 days, with a range of 31–60 days. This indicates that traditional genotypes initiated storage roots 9.1 days later than improved genotypes (Figure 7).

### 2.2. Genetic Gain

Traditional sweet potato cultivars used in Cuba before 1970 had an average yield of 9.60 t ha^−1^. A slight increase in yield (3.93 t ha^−1^) was observed in the decade of 1971–1980, which reached 13.53 t ha^−1^. A considerable increase in yield was obtained with the improved genotypes of the decade of 1982–1990, which had an average yield of 23.93 t ha^−1^. Compared with the traditional genotypes, the increase was 14.33 t ha^−1^. In the decades of 1991–2000 and 2001–2010, there was no considerable increase in yield compared with the decade of 1981–1990. It was only in the decade of 2011–2020 that there was a significant increase in this characteristic (34.22 t ha^−1^) compared with the previous decades. Therefore, it can be said that for the improved genotypes of the last decade (2011–2020), compared with the genotypes before starting the genetic improvement program (traditional genotypes), the yield has had an improvement progress of 24.62 t ha^−1^, meaning that the yield in Cuba has increased by 256% in the last 50 years due to plant breeding (Figure 8).

The accumulated genetic gain in the different decades showed different genetic progress in terms of yield increase of the improved genotypes compared with the traditional ones. In the 1971–1980 decade, the genetic gain was 0.20 t ha^−1^ year^−1^, indicating moderate yield progress. After 1981–1990, the genetic gain was 0.37 t ha^−1^ year^−1^, showing significant progress in genetic improvement. Then, in the 1991–2000 and 2001–2010 decades, it was 0.28 and 0.22 t ha^−1^ year^−1^, respectively. Finally, in the 2011–2020 decade, the genetic gain was 0.26 t ha^−1^ year^−1^, which reflects a sustained genetic progress over the years. This shows that genetic improvement in Cuba has achieved a constant increase in sweet potato performance, with an annual genetic gain that varies according to the decade and the intensity of selection applied (Figure 9).

The tuberous roots of traditional sweet potato cultivars used before 1970 were almost circular in shape, with an average circularity of 0.85. Furthermore, all tuberous roots exhibited root defects (longitudinal grooves, horizontal constrictions and veins). During the next three decades (1971–1980, 1981–1990 and 1991–2000), genetic improvement made no progress in this regard, as the improved genotypes resulting from these decades exhibited the same defects on the surface of the tuberous roots. It was not until the decade 2001–2010 (2005 to be precise) that the first improved genotype with smooth skin and no surface defects was obtained. In addition, the circularity values for tuberous roots shape decreased over time, indicating that the current improved cultivars have sweet potatoes that are less circular and elliptical in shape. The average circularity of improved cultivars in the last decade is 0.66 (Figure 10A).

All traditional cultivars planted before 1970 in Cuba had a creamy-white flesh color of their tuberous roots, with an average lightness value of 87.21. During the next three decades (1971–1980, 1981–1990 and 1991–2000), the improved and released cultivars had white flesh of their tuberous roots, similar to the traditional cultivars. It was only in the decade of 2001–2010 that cultivars with yellow (2005) and orange (2006) flesh were obtained for the first time (commercial scale). In the last decade, all the newly released cultivars had orange or purple flesh color. The first cultivar with purple flesh was released in 2011. The average luminosity in the decade 2011–2020 was 64.63, which shows that the improvement for flesh color has been in the direction of obtaining cultivars with more intense pigmentation (Figure 10B).

### 2.3. Tolerance/Susceptibility to Weevil (Cylas formicarius Fab.) and Tolerance/Susceptibility to Nematode (Meloidogyne incognita [Kofoid & White] Chitwood)

Based on the simultaneous selection for several traits (yield and percentage of infestation by the weevil [*C. formicarius*]) using independent criteria for the 20 sweet potato cultivars (improved and traditional), seven cultivars (five improved and two traditional) showed less than 20% weevil infestation. Six genotypes were positively selected (red shaded area), five of which are improved and one of which is traditional. This shows progress in tolerance to this pest due to the breeding program. Interestingly, two traditional cultivars demonstrated consistent resistance to the weevil. This phenotypic resilience is likely due to long-term selection pressure exerted by farmers, as these landraces have been cultivated in weevil-endemic regions of Cuba since the early 20th century (Figure 11A).

On the other hand, regarding the reproduction factor for the nematode (*M. incognita*), one traditional cultivar was found to be an excellent host (FR > 10) and eleven cultivars were found to be good hosts (1 < FR < 10) (six traditional and five improved), while eight cultivars were found to be non-hosts (FR < 1) (five improved and three tolerant). Based on simultaneous selection on several traits (yield and nematode reproduction factor), five cultivars were selected (four improved and one traditional), so there has been genetic progress for tolerance to *M. incognita* (Figure 11B).

### 2.4. Tolerance/Susceptibility to Viral Disease and Drought Tolerance/Susceptibility

Improved and traditional cultivars showed different incidence rates of the viral disease in sweet potato. Five cultivars (three traditional and two improved) had incidence rates greater than 81%, two traditional cultivars had incidence rates between 41 and 80%, six cultivars (four traditional and two improved) had incidence rates between 11 and 40%, and seven cultivars (one traditional and six improved) had incidence rates below 10%. Of the ten traditional cultivars, nine (90%) had incidence rates above 10%, while of the ten improved cultivars, six (60%) had incidence rates below 10%. Therefore, the tolerance/resistance level of improved cultivars is higher than that of the traditional cultivars (Figure 12A).

Yield loss under water deficit conditions and the relative variation in yield potential under normal moisture and water deficit conditions of the traditional and improved cultivars resulted in different responses. Of the 20 cultivars evaluated, only four improved cultivars had Stress Susceptibility Index (SSI) values below 0.8 (drought tolerant), and all traditional cultivars had values above 1 (drought susceptible). The average SSI value for traditional cultivars was 1.44, and for improved cultivars, it was 0.85. Mean yield (MP) expresses the average yield produced between the irrigated and non-irrigated treatments for the same cultivar. Of the ten improved cultivars, nine had an MP greater than 20 t ha^−1^, while all the traditional cultivars had MP values less than 20 t ha^−1^. This demonstrates that the improved cultivars have a high degree of improvement in drought tolerance, while the traditional cultivars are susceptible (Figure 12B).

### 2.5. Phenotypic Stability

The genotypes (traditional and improved) and the environments (spring and winter) were plotted in a GGE biplot. All genotypes (except one) were below the horizontal center line, suggesting higher yields in winter and lower yields in spring. Nine improved genotypes had an angle between the genotype vector and the environment vector (spring) of less than 90° (acute angle), indicating that they yielded more than the average of all genotypes in spring. The traditional genotypes and one improved genotype were below the average in spring (obtuse angles). Furthermore, the group of traditional genotypes was located opposite the spring season, suggesting that this season is not suitable for planting this type of genotype (Figure 13A,B).

To determine which cultivars were best adapted (highest yield) to each environment (spring and winter), straight lines were projected through the origin of the biplot and the environments, and the genotypes were classified according to their projections on the axis in the direction of the environment. Thus, in both spring and winter, the genotypes with the highest yield were the improved ones, while those with the lowest yield were the traditional ones. The straight line passing through the origin of the biplot and perpendicular to the axes (spring and winter) divides the genotypes into two groups: those with above-average yields and those with below-average yields. In spring, only eight genotypes showed above-average yields; all of them improved, while in winter, eleven genotypes showed above-average yields (eight improved and three traditional). Furthermore, the longer lengths of the genotype vector linking the biplot origin to the environmental marker suggest that the spring season has a greater ability to discriminate between genotypes; the shorter vectors in the winter season for some improved genotypes implied that they tended to have similar yield in the associated environment (Figure 14A,B).

To compare some traditional varieties with improved varieties, a line was drawn connecting them and another perpendicular to it passing through the origin of the biplot. The two environments (spring and winter) are located to the right of the perpendicular, as are most of the improved cultivars (8–9), indicating that the improved cultivars (80–90%) had higher yields than the traditional cultivars in both environments (Figure 15).

A polygon (“convex hull”) was constructed by joining the cultivars furthest from the biplot origin (vertex cultivars). These cultivars are those with longer vectors in their respective directions, which measures their responsiveness to the environments. Therefore, the vertex cultivars were the most sensitive, that is, the best or worst in some or all of the test environments. It was observed that seven improved varieties are in the same sector (Sector 1), separated from the rest of the biplot by two perpendicular lines. This confirms that the improved varieties had higher yields in both seasons. The improved vertex variety (‘INIVIT B-50’) in Sector 1 was the highest, yielding in both environments. On the other hand, the traditional cultivars and the three improved cultivars had no environments in their sectors, indicating that they were not the best in any of the test environments; in fact, they were the lowest yielding in one or both of the test environments (Figure 16A).

The proximity of eight improved genotypes (to the right of the perpendicular) and the acute angles they form with respect to the origin indicate a similarity or positive relationship between them. On the other hand, the environments (spring and winter), despite being in the same sector, cannot be grouped within the same mega-environment because they form an obtuse angle between them, suggesting little association between the environments. The improved genotypes are (on average) closer to the ideal environment (concentric circles) (Figure 16B).

To evaluate the average yield and stability of the genotypes, the average environment coordinate (AEC) was plotted. This represents the average environment obtained as the mean of the environment coordinates (small blue circle, Figure 17A). A straight line was drawn through the origin of the biplot and the average environment, called the abscissa axis of the AEC. The projections of the varieties on this axis represent the average yield of the varieties, and these are arranged along the axis with the arrow pointing to the highest yield. This indicates that eight improved varieties (80%) had the highest average yield in the average environment; furthermore, these eight had little projection on the ordinate axis, which indicates high stability. However, one improved genotype had the greatest projection on the ordinate axis (a point above the abscissa axis), indicating its instability. In addition, traditional varieties are located on the left side of the y-axis, indicating low yield and instability in the studied environments (Figure 17A).

The ideal genotype does not normally exist, but this analysis was nonetheless used as a reference in the evaluation of traditional and improved varieties. The ideal genotype must simultaneously have the highest average yield and high stability in all environments. The concentric circles (AEA: absolutely stable, Figure 17B) centered on the ideal genotype allowed for visualization of the distance between all genotypes and the ideal genotype. Seven improved genotypes had the smallest distance from the ideal genotype, which is a measure of their desirability. One of the improved varieties (’INIVIT B-50’) is closest to the ideal genotype and is therefore the most desirable of all the cultivars evaluated.

## 3. Discussion

The results showed that the improved sweet potato cultivars significantly outperformed the traditional cultivars in tuberous root yield, with an increase of 17.11 t ha^−1^. This increase is consistent with recent studies highlighting the impact of genetic improvement in sweet potato, such as those reported to increase yield by 30–50%, depending on environmental conditions [18]. Furthermore, the greater precocity of storage root initiation in improved cultivars (33.06 days versus 42.16 days in the traditional cultivars) suggests improved adaptation to variable environmental conditions, as precocity is a key trait for adaptation to climate change in tropical crops [19]. Genetic improvement in sweet potato has not only increased yield but also improved nutritional quality, especially carotenoids and anthocyanin content, which is relevant for food security [8,20]. Genetic gain analysis showed a progressive increase in the yield of sweet potato varieties from the 1970s to the present, with an increase of 256% in the last 50 years. Genetic gain ranged between 0.20 and 0.37 t ha^−1^ year^−1^ for decades, demonstrating that genetic improvement of sweet potato in Cuba has achieved a constant increase in the yield of this crop. This progress is comparable with documented results of yield improvement in crops such as corn and rice through conventional genetic improvement [21].

Furthermore, the reduction of surface defects in tuberous roots, the reduction of roundness, and the improvement of the flesh color (from white to orange and purple) reflects advances in the selection of desirable market traits and high nutritional quality, particularly β-carotene (provitamin A) and anthocyanins, which are associated with reduced vitamin A deficiency and antioxidant activity [8].

The improved varieties showed lower weevil infestation, with five improved genotypes selected for their resistance. This finding is consistent with recent studies indicating that genetic improvement of sweet potato can reduce susceptibility to the weevil and also reduce yield losses [22], especially in regions where the weevil is a limiting pest [23]. The use of genotypes with high yield and low percentage of weevil infestation suggests that genetic improvement can contribute to more sustainable management of this pest.

Regarding nematode tolerance, the improved varieties showed a lower reproduction factor (RF), with eight genotypes classified as non-host (RF < 1). Selecting sweet potato genotypes with low host capacity is important to reduce nematode populations in the soil [24]. Furthermore, nematode resistance is a key trait in sweet potato breeding, as these pathogens can cause significant yield losses [18,25]. Nematode resistance in improved sweet potato varieties is also combined with other agronomic traits, such as yield and nutritional quality, maximizing the benefits of genetic improvement.

Improved varieties showed lower incidence of viral diseases, with 60% of the improved genotypes presenting incidence rates below 10%. This result is similar to others reporting that sweet potato breeding can increase resistance to viral diseases in sweet potato [18,26]. Therefore, the selection of virus-resistant genotypes can improve the productivity and sustainability of agricultural systems. Improved varieties showed higher drought tolerance, with an SSI lower than 0.8, compared with traditional varieties (SSI > 1). These tolerant varieties can express several resistance mechanisms including escape, evasion, tolerance or recovery from water stress [27]. Therefore, the presence of one or more of these drought resistance mechanisms in diverse combinations can provide a variety that is tolerant to drought at a given stage of its growth cycle. Drought-tolerant sweet potato varieties that produce high yields in the absence of sufficient rainfall or irrigation compared with other varieties have been reported [27]. Consequently, drought-tolerant genotypes can improve yield stability under adverse climatic conditions.

Phenotypic stability analysis using a GGE biplot confirmed that improved cultivars had higher yield and stability under different environmental conditions (spring and winter). A recent study evaluated the phenotypic stability of 15 sweet potato cultivars across multiple locations in sub-Saharan Africa, which showed that two sweet potato cultivars had phenotypic stability and maintained consistent yield despite variations in climatic and soil conditions [28]. On the other hand, Grüneberg et al. [14] evaluated 1174 sweet potato clones across five contrasting environments in Peru, revealing significant genetic variation in storage root yield (0–55.5 t/ha) and adaptability. While altitude effects were not explicitly analyzed, genotypes like Xuzhou 18 demonstrated broad stability, whereas others (e.g., Tanzania, Resisto) showed divergent responses to water stress. The study emphasizes the role of Harvest Index (HI) and genotype-by-environment interaction (σ^2^G × E) in selecting context-specific varieties.

This study highlights the importance of genotype–environment (GxE) interactions in determining yield and suggests that cultivar selection should consider not only yield potential but also the ability to maintain that yield in different environments. However, phenotypic stability does not always correlate with high yields. A study in the South Pacific identified cultivars that had high phenotypic stability but did not achieve the maximum yields observed in other less stable cultivars [29]. Phenotypic stability in sweet potato is complex, suggesting that breeders may face a trade-off between stability and maximum yield. Furthermore, there is no direct relationship between stability and maximum yield, suggesting the need for a balanced approach in breeding programs. Although our program was based on phenotypic selection, future integration of genomic tools (e.g., RAD-seq) could identify QTL associated with phenotypic stability, accelerating the selection of resilient genotypes. This would complement Cuba’s successful phenotypic approach while overcoming the challenges of hexaploidy.

The advances of the sweet potato improvement program in Cuba with cultivars tolerant to drought (SSI < 0.8), pest resistance and phenotypic stability provides a key model for food security in the face of climate change. These cultivars will not only maintain productivity in Cuban regions with erratic rainfall but could also be scaled up to other vulnerable tropical areas, such as the insular Caribbean or dry areas of Central and Northern South America, where sweet potato is an essential crop for smallholder farmers. The combination of high productivity (up to 34.2 t ha^−1^) and resilience to biotic/abiotic stresses achieved without molecular tools demonstrates that massive phenotypic selection strategies, such as those developed at INIVIT, are replicable in resource-limited contexts. To facilitate their regional adoption, future studies should evaluate the performance of these genotypes in multi-site networks, prioritizing areas with marginal soils and high climatic variability, similar to Cuban conditions.

## 4. Materials and Methods

### 4.1. Study Area

The study was conducted at the Research Institute of Tropical Roots and Tuber Crops (INIVIT) in Santo Domingo, Villa Clara, Cuba (22°35′00″ N, 80°14′18″ W; 50 m above sea level), during 2023 and 2024. The site has a calcareous brown soil [30].

Meteorological data were recorded at the automatic meteorological station belonging to the institute (national meteorological network code: 78326, data: http://www.insmet.cu [accessed on 8 June 2024]).

### 4.2. General Experimental Design

All field experiments shared the following common design elements: The planting material consisted of apical stem cuttings (30 cm in length, with 4 buried internodes). The plot layout comprised 5 rows of 5 m in length (effective area of 22.5 m^2^), with a planting density of 0.90 × 0.30 m spacing (37,037 plants/ha). The experimental units followed a completely randomized block design, with 5 replications for Experiments 1–2 and 3 replications for Experiments 3–7. Border plants (outer rows) were excluded from evaluations. Irrigation was provided via a sprinkler system (250 m^3^/ha weekly), with no application of chemical or organic fertilizers, following standard agronomic practices according to INIVIT technical guidelines [31].

### 4.3. Progress in the Genetic Improvement of Some Traits (Experiment # 1)

To determine whether there has been progress in the genetic improvement of sweet potatoes in some traits, 10 traditional varieties (the most important in Cuba before 1972, before the start of the genetic improvement program) and 10 improved cultivars obtained and released by the genetic improvement program of INIVIT (the most important at the national level) were selected. The names of the varieties and the year of release are shown in Table 2.

Two plantations were carried out in November 2023 (winter season) and May 2024 (spring season) (Figure 18).

#### 4.3.1. Variables Evaluated

##### Tuberous Root Yield

Harvesting took place 120 days after planting. The tuberous roots in the three central furrows were weighed, excluding the plants at the edges. Weighing was performed with a 50 kg digital hanging scale, Mod 16730-2 TorRey.

##### Commercial Tuberous Root Yield

The measurements were performed in the same way as above, but only tuberous roots weighing more than 80 g were included, as this threshold aligns with Cuba’s national marketing standards for commercial sweet potato production.

##### Number of Tuberous Roots per Plant

The total number of tuberous roots in the three central furrows of each plot was counted and divided by the number of plants at the time of harvest.

##### Foliage Yield

The determination was carried out 90 days after planting. A square metal frame was made, with a side length of 50 cm, for an area of 2500 cm^2^. This metal frame was randomly placed in the middle of each plot, where two measurements were created per plot for each cultivar and the foliage inside the frame was cut and immediately weighed. This measurement was performed between 0900 and 1000 h.

##### Dry Matter of Tuberous Roots

Ten samples of approximately 100 g fresh weight were taken from each cultivar. Each sample was divided into small parts and dried in a forced-circulation oven (TSN 50) at 65 °C for five days until a constant weight was reached.(1)$$\%Drymatter=\frac{Dry\;weight}{Wet\;weight}\;*\;100$$

##### Total Soluble Solids of Tuberous Roots (°Brix)

Total soluble solids were determined as °Brix using a Hanna digital refractometer (Hanna instruments, Woonsocket, R) with a range of 0 to 85% with a precision of 0.1 °Brix. The evaluation was carried out at the time of harvest. Ten samples of tuberous roots were taken per genotype, and three readings were taken from each sample.

##### Storage Root Initiation

To determine the storage root initiation after 15 days of planting, daily sampling was carried out in the experimental plots, defining tuberous roots as those elliptical roots thickened in the middle section with a dimension greater than 0.6 cm in diameter.

### 4.4. Genetic Gain (Experiment # 2)

To determine the genetic progress over the years, the 24 sweet potato cultivars were selected, four for each evaluation period (six periods). The names of cultivars and the year of release are presented in Table 3.

Two plantations were carried out in November 2023 (winter season) and May 2024 (spring season). The genetic gain was calculated from the average yield of the two planting seasons using the following formula:(2)$$\Delta G=\frac{i\;*\;h^{2}\;*\;\sigma_{p}}{T}$$
where:

i = Selection intensity

h^2^ = Narrow-sense heritability

σ_p_ = Phenotypic standard deviation of yield

T = Time interval between generations

Narrow-sense heritability (h^2^) was estimated from parent–offspring regressions in segregating populations from the INIVIT program using historical performance data from 20 selection cycles. Additive genetic variance (σ^2^_a_) was calculated by ANOVA tests, considering the adjusted means of half-sib families.

The selection intensity (i) was calculated theoretically assuming 5% of selected individuals (i = 2.06), which is the standard proportion used in the INIVIT program for massive segregating populations.

#### Shape and Color of Tuberous Roots

At each harvest (winter and spring seasons), the samples were photographed individually with a Canon EOS 600D camera (Fukushima Canon Inc., Fukushima, Japan). The tuberous roots of each cultivar were washed and dried before determining their color. Two morphometric variables were used: one related to the shape and dimensions of the tuberous root (Circularity: C) and another related to the color of the tuberous root flesh (Luminosity: L*). Three points on the flesh (cross-section at the midpoint) were measured in three tuberous roots per genotype, for a total of nine data points per cultivar. In addition, a controlled environment, in terms of temperature (20 ± 2 °C), relative humidity (75 ± 5%) and illumination (500 lx), was maintained during the measurement process. The professional digital image analysis software programmed in Java was used: ImageJ ver. 1.46 from the National Institute of Health. The color space used was CIE L*a*b* from 1976 of the International Commission on Illumination [32].

Circularity:(3)$$C=\frac{4\pi \;*\;\left(Area\right)}{{Perimeter}^{2}}$$

### 4.5. Tolerance/Susceptibility to Weevil (Cylas formicarius Fab.) (Experiment # 3)

The experimental trials were established in November 2023 and September 2024, with two replicate plantings per season conducted in spatially separated fields (1500 m apart) at INIVIT. The 20 cultivars used in Experiment 1 were evaluated (10 traditional cultivars and 10 improved cultivars).

The experimental design was based on that proposed by Talekar [33], which consisted of planting two 0.90 m wide border furrows with a distance of 2.70 m between them, leaving two empty furrows in the middle of these two rows. These two border furrows were planted two months before the planting of the 20 varieties to be evaluated. These two border furrows were planted with a susceptible cultivar ’Cautillo’ to create a source of infestation inoculum before planting the materials to be evaluated. Rotary tilling was carried out on the two central furrows, and the 20 cultivars to be evaluated were planted. For each cultivar, two-row plots were used, 0.90 m wide and 5 m long, with an area per plot of 9 m^2^, with three replicates per cultivar in each test. Using this method, each cultivar to be evaluated is located in the center of the infested inoculum source, which ensures a uniform infestation.

Harvesting was carried out 150 days after planting. All the roots harvested by cultivar were collected individually. The percentage of infestation per cultivar was determined:(4)$$\%\;infestation=\frac{Weight\;of\;infested\;roots}{Total\;weight\;of\;roots}\;*\;100$$

A simultaneous selection was performed on two traits (yield and % infestation) with independent criteria. Selection criteria were established for each trait, with minimum yield set at 15 t ha^−1^ and maximum infestation threshold at 20%.

### 4.6. Tolerance/Susceptibility to Nematode (Meloidogyne incognita [Kofoid & White] Chitwood) (Experiment # 4)

The inoculum was prepared using tomato (*Solanum lycopersicum* L.) roots with a pure population of *Meloidogyne incognita* from sweet potato. Inoculum purity was confirmed by morphological identification of J_2_ (160× stereoscopic) and molecular analysis (PCR-ITS), compared with type sequences in GenBank. The population was isolated in Rutgers tomato cultivars to avoid contamination.

Pots with 1 kg capacity were used containing a mixture of carbonated loose brown soil (pH 6.8–7.2, sandy loam texture), and cattle manure was placed as organic fertilizer (in a 1:1 ratio), previously sterilized by steam at 121 °C for 2 h. The 20 cultivars of sweet potato used in Experiment 1 were evaluated. One in vitro plant (with 4–6 leaves and 6 cm in height) was planted per pot, and a period of 15 days was waited for inoculation. The inoculum (second-stage juveniles [J_2_]) was obtained using the modified Hemming and Whitehead tray method, and the inoculum was introduced into the soil through four holes on the root system and in the area close to the stem, following the methodology of Coyne and Ross [34]. The initial population level used was 1.0 J_2_ g^−1^ soil (equivalent to 1000 juveniles per pot).

The plants were kept in the biological isolator for 100 days and were irrigated on alternate days. After this period, the plants were extracted and transferred to the laboratory for processing.

The final population was established by adding the population extracted from the roots, and specimens in the soil. The latter were taken from three subsamples (100 g each) of the soil from each replicate, which were processed using the modified Hemming and Whitehead tray method [34]. The solutions were quantified by direct counting of the J_2_ in a Zeiss^®^ stereoscopic microscope (Carl Zeiss, Jena, Germany) at 160×.

The reproduction factor was determined:(5)$$RF=\frac{Final\;population}{Initial\;population}$$

To establish the capacity categorization as a host, the scale of Ferris et al. [35] was used:

FR ≥ 10: excellent host

1 < FR < 10: good host

FR = 1: maintenance host

0 < FR < 1: poor or non-host

A simultaneous selection was performed on two traits (reproductive factor and yield) with independent criteria. For each tray, a value was set as a selection criterion: a minimum value of 15 t ha^−1^ for yield and a maximum value of 1 for the reproduction factor.

### 4.7. Tolerance/Susceptibility to Viral Disease (Experiment # 5)

Experimental plots were established in November 2023, with two spatially separated field trials (1500 m apart) at INIVIT The 20 cultivars used in Experiment 1 were evaluated (10 traditional cultivars and 10 improved cultivars).

The experimental design was the same as that used in Experiment 3. The cultivar used in the border furrows susceptible to disease was ‘CEMSA 90-510’ to create a source of infestation inoculum before planting the materials to be evaluated.

The presence of sweet potato virus disease (SPVD), a viral complex affecting sweet potato crops, was evaluated. Characteristic symptoms included stunted growth, chlorotic leaf mottling, leaf deformation, and reduced plant vigor. These manifestations were visually recorded in each plant throughout the crop cycle. The harvest was carried out 120 days after planting. The percentage of incidence of the viral disease was determined:(6)$$\%\;Incidence=\frac{Plants\;with\;symptoms}{Total\;plants}\;*\;100$$

Virus and nematode symptoms were scored independently by three trained technicians blinded to cultivar identity.

### 4.8. Drought Tolerance/Susceptibility (Experiment # 6)

The experiment was planted in November 2023. Precipitation and relative humidity were recorded at the experimental site (Table 4).

Four irrigated blocks (Treatment 1) and four non-irrigated blocks (Treatment 2) were planted, with each block containing the 20 cultivars used in Experiment 1, as well as the plot design and size. Sprinkler irrigation was applied to all blocks for the first three weeks until all plants were established, and then irrigation was suspended for Treatment 2 to induce a period of water stress, but irrigation was continued for Treatment 1 throughout the period until harvest at 120 days.

At harvest, the two outer rows of each plot were discarded, and only the three inner rows with 45 evaluable plants were used. For the evaluation of tuberous root yield, the total number of roots per net plot was counted and weighed.

To evaluate the tolerance/susceptibility of the varieties to drought, two stress tolerance indices were used based on their performance under normal irrigation and water deficit conditions. The drought tolerance indices were calculated as follows:

Stress Susceptibility Index [36](7)$$SSI=\frac{1-\left(\frac{Ys}{Yp}\right)}{1-\left(\frac{Yms}{Ymp}\right)}$$

Mean Productivity [37](8)$$MP=\frac{Yp+Ys}{2}$$
where:

Yp: Yield of a genotype in normal irrigation condition

Ys: Yield of a genotype in water deficit condition

Ymp: Mean yield in normal irrigation condition

Yms: Mean yield in water deficit condition

A simultaneous selection was performed on two traits (Stress Susceptibility Index and Mean Productivity) with independent criteria. For each tray, a value was set as a selection criterion, with a minimum value of 20 for Mean Productivity and a maximum value of 0.80 for Stress Susceptibility Index.

### 4.9. Phenotypic Stability (Experiment # 7)

To estimate the phenotypic stability, the combination of the two seasons of the year (winter and spring) for two years (2023 and 2024) was defined as the environment and the 20 cultivars indicated in Experiment 1 were defined as genotypes.

The multivariate model used was SREG: Sites Regression Model, using GGE biplot (Genotype + Genotype-by-Environment), available on the website of CIMMYT (Centro Internacional de Mejoramiento de Maíz y Trigo) in the Biometrics and Statistics Unit (BSU) or at http://www.cimmyt.org/english/wps/biometrics/ (accessed on 12 December 2024).

The GGE biplot model was selected for phenotypic stability analysis due to its capacity to: decompose G × E interaction into interpretable components (genotype main effect + G×E interaction), excluding environmental noise; and visualize complex response patterns through biplot graphics, enabling simultaneous evaluation of: genotype performance, stability across environments and mega-environment identification.

The biplot construction followed Yan et al. (2020) [19], based on an environment-centered principal component analysis of standardized yield data. The biplot is constructed from the first two components of a principal component analysis, using the regression model by sites (SREG), according to the following linear model:Y_ij_ − ȳ_.j_ = (λ_1_ξ_i1_η_j1_) + (λ_2_ξ_i2_η_j2_)+ ε_ij_(9)
where:

Y_ij_: average yield of genotype i in environment j

ȳ_.j_: mean of the genotypes in environment j (marginal mean)

λ_1_ξ_i1_η_j1_: first principal component (Pc1)

λ_2_ξ_i2_η_j2_: second principal component (Pc2)

λ_1_ y λ_2_: eigenvalues associated with Pc1 and Pc2, respectively

ξ_i1_ y ξ_i2_: eigenvectors of genotypes i for the first and second principal components, respectively

η_j1_ y η_j2_: eigenvectors of the j environments for the first and second principal components, respectively

ε_ij_: experimental error associated with genotype i in environment j

### 4.10. Statistical Analysis and Visualization

Statistical analyses were performed using RStudio (v2023.03.1) with the following approach: First, a two-way ANOVA test was conducted to assess the effects of cultivars (traditional vs. improved) and growing season (winter vs. spring) on all agronomic traits (yield, commercial yield, tuber number, foliage yield, dry matter, and °Brix). Data met all ANOVA assumptions, as confirmed by normality (Shapiro–Wilk test, *p* > 0.05 for all variables) and homoscedasticity (Levene’s test, *p* > 0.05). When the ANOVA test detected significant differences (*p* < 0.05), post hoc comparisons were performed using Fisher’s LSD test. All visualizations were generated with ggplot2 (v3.4.2) [38].

## 5. Conclusions

This study demonstrates that Cuba’s sustained sweet potato breeding program has successfully overcome the challenges of hexaploid genetics through phenotypic selection, achieving remarkable genetic gains despite resource limitations. Over five decades, the program developed improved varieties that outperform traditional cultivars by 256% in yield (34.22 vs. 9.60 t ha^−1^) while exhibiting superior stress resilience including drought tolerance (SSI < 0.8), reduced weevil infestation (60% decrease), and nematode resistance (40% of varieties). The flagship cultivar ‘INIVIT B-50’ exemplifies these advances, showing exceptional yield stability across diverse environments. These achievements, now implemented across 98% of Cuba’s sweet potato area, provide both a practical model for tropical root crop improvement and a foundation for climate-resilient food security. The program’s success underscores the potential of long-term phenotypic selection for complex polyploid crops, particularly in resource-constrained agricultural systems.

## 6. Patents

The 10 improved sweet potato cultivars are registered in the official list of commercial varieties.

## Figures and Tables

**Figure 1 plants-14-01911-f001:**
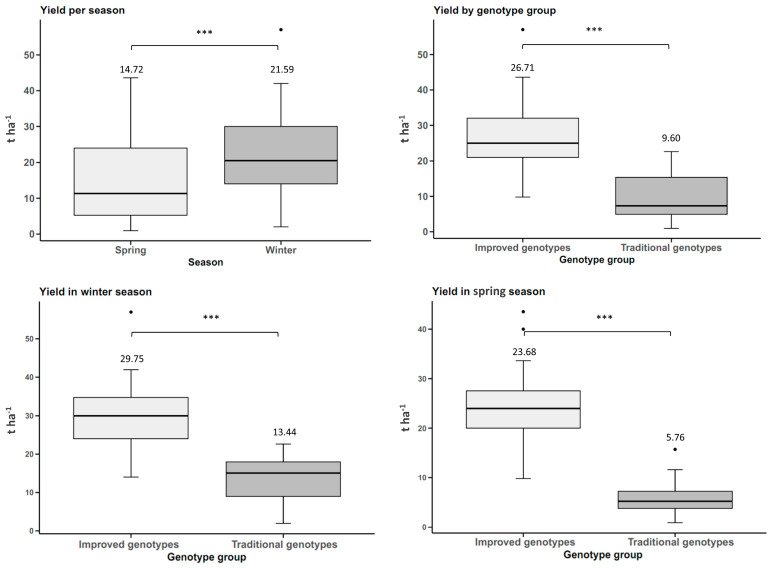
Yield of sweet potato tuberous roots (n = 20 cultivars × 5 replicates) by planting season (winter and spring) and genotype group (improved and traditional). *** *p* < 0.001.

**Figure 2 plants-14-01911-f002:**
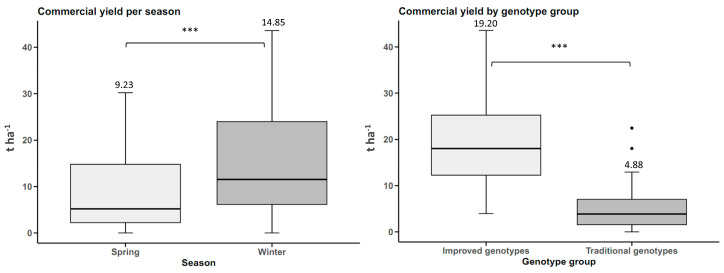
Yield of commercial sweet potato tuberous roots (n = 20 cultivars × 5 replicates) by planting season and by genotype group (improved and traditional). *** *p* < 0.001.

**Figure 3 plants-14-01911-f003:**
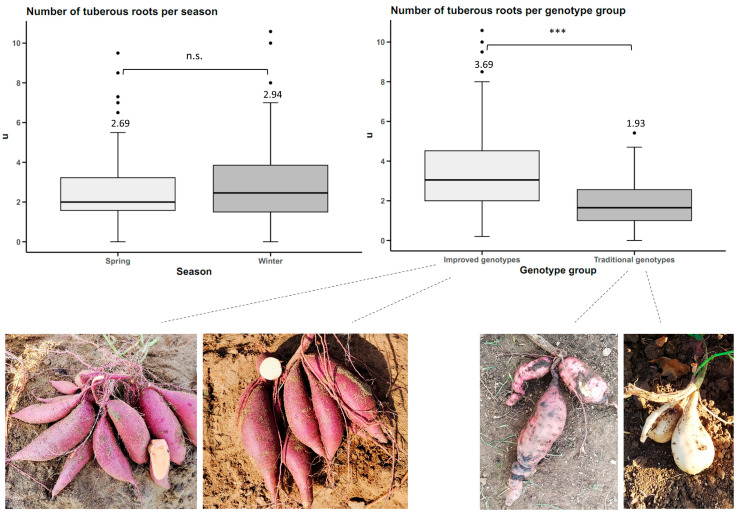
Number of sweet potato tuberous roots (n = 20 cultivars × 5 replicates) by planting season and by genotype group. n.s.: not significant; *** *p* < 0.001.

**Figure 4 plants-14-01911-f004:**
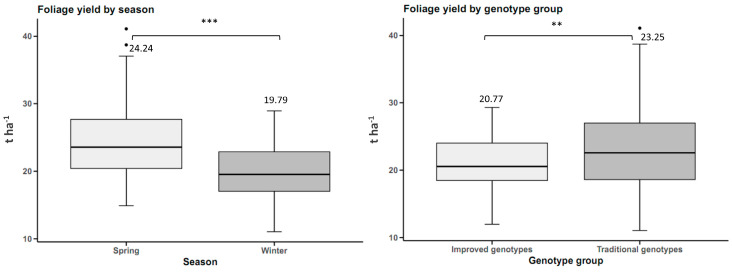
Foliage yield in sweet potato (n = 20 cultivars × 5 replicates) by planting season and by genotype group. n.s.: not significant; ** *p* < 0.01, *** *p* < 0.001.

**Figure 5 plants-14-01911-f005:**
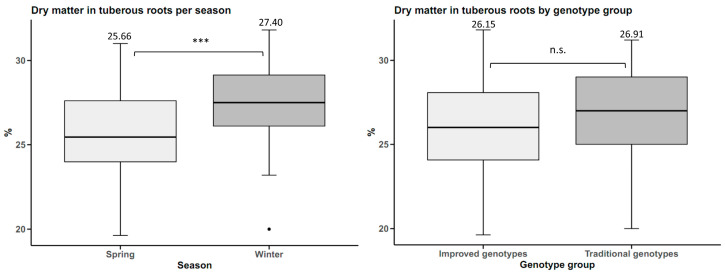
Dry matter yield in sweet potato (n = 20 cultivars × 5 replicates) by planting season and by genotype group. n.s.: not significant; *** *p* < 0.001.

**Figure 6 plants-14-01911-f006:**
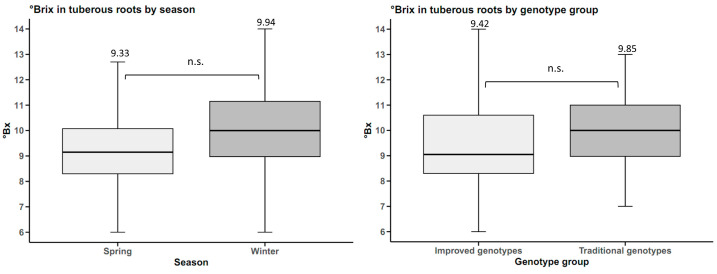
°Brix in sweet potato tuberous roots (n = 20 cultivars × 5 replicates) by planting season and by genotype group. n.s.: not significant.

**Figure 7 plants-14-01911-f007:**
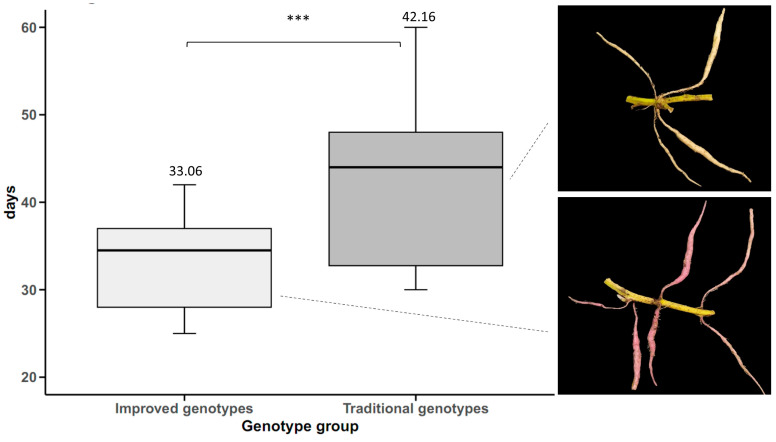
Storage roots initiation in sweet potato (n = 20 cultivars × 5 replicates) by genotype groups. *** *p* < 0.001.

**Figure 8 plants-14-01911-f008:**
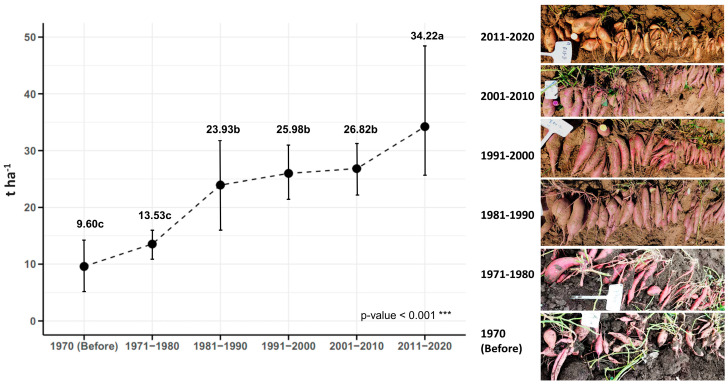
Sweet potato tuberous roots yield (n = 24 cultivars × 5 replicates) over time, by decades, based on the year of release of the genotypes. Different letters (a, b, c) indicate statistically significant differences (*p* < 0.001).

**Figure 9 plants-14-01911-f009:**
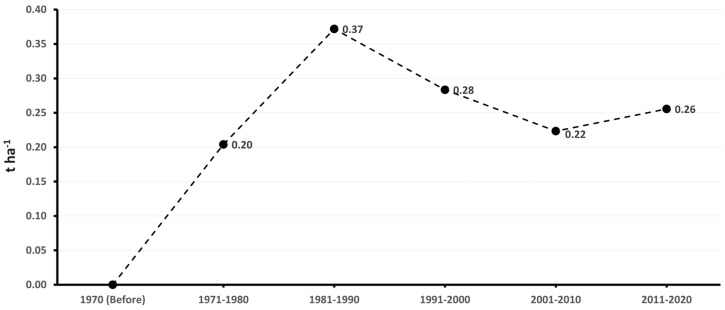
Accumulated genetic gain of the yield of sweet potato cultivars in the different decades.

**Figure 10 plants-14-01911-f010:**
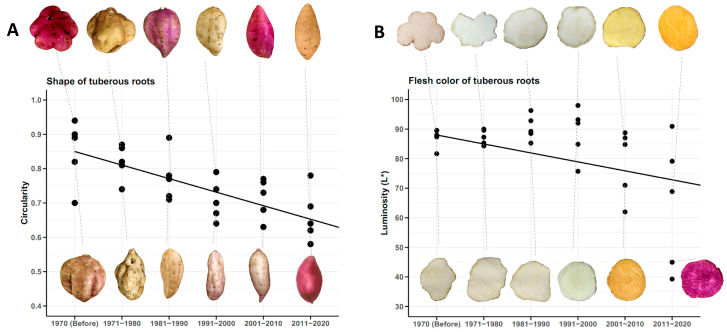
Shape of tuberous roots (n = 24 cultivars × 3 replicates) expressed in circularity (**A**), and flesh color (**B**), by decades, depending on the year of release of the genotypes.

**Figure 11 plants-14-01911-f011:**
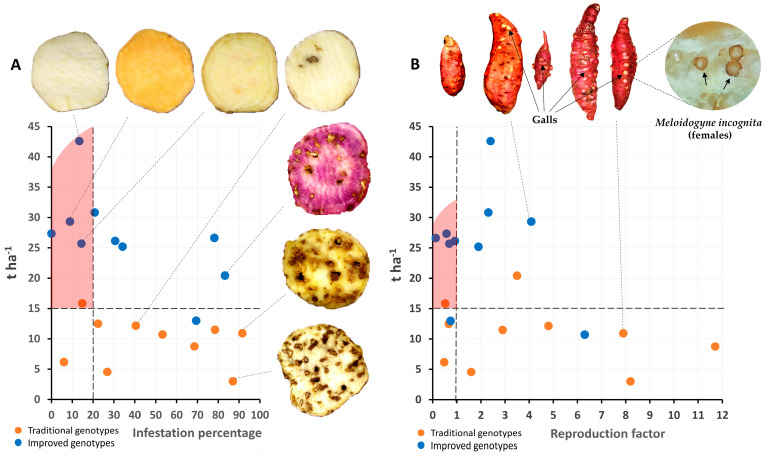
Selection of genotypes (n = 20 genotypes × 3 replicates) according to independent criteria (yield and percentage of infestation by weevil) (**A**) and selection of genotypes according to independent criteria (yield and reproduction factor of *M. incognita*) (**B**).

**Figure 12 plants-14-01911-f012:**
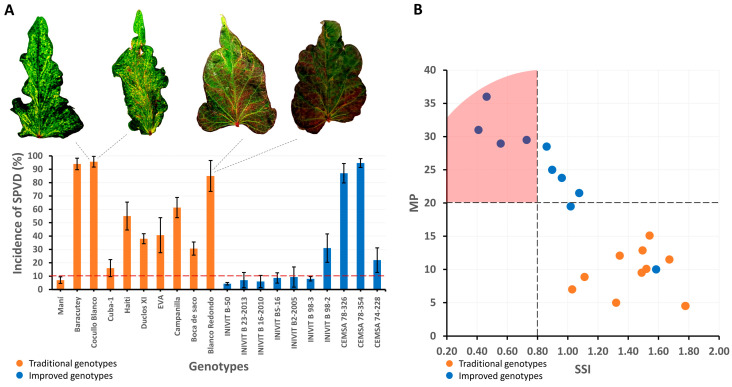
Incidence of sweet potato viral disease in 20 sweet potato genotypes (traditional and improved) (n = 20 genotypes × 3 replicates) (**A**) and selection of genotypes according to independent criteria with drought tolerance indices (MP and SSI) (**B**).

**Figure 13 plants-14-01911-f013:**
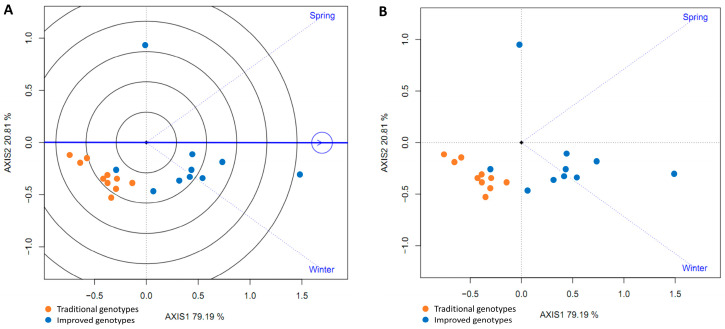
GGE biplot based on yield data from 20 sweet potato cultivars (traditional and improved) at two seasons (**A**,**B**) (4 replicates).

**Figure 14 plants-14-01911-f014:**
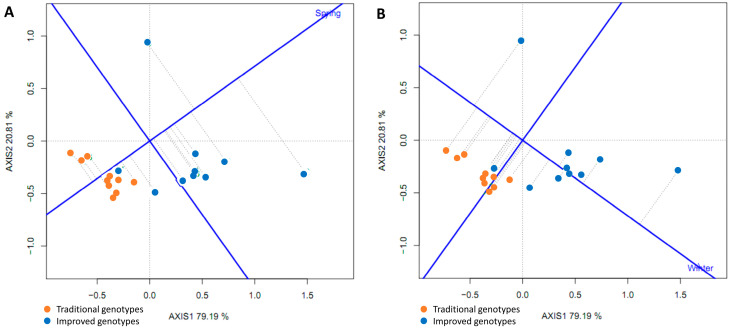
GGE biplot based on yield data from 20 sweet potato cultivars (traditional and improved) and two environments: spring (**A**) and winter (**B**) (4 replicates).

**Figure 15 plants-14-01911-f015:**
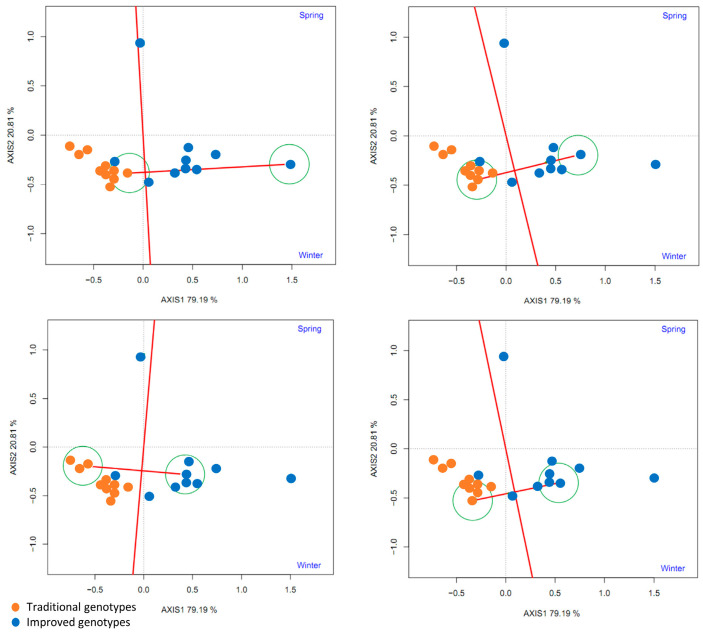
GGE biplot based on the comparison of the performance of four traditional and improved varieties and two environments (spring and winter) (4 replicates).

**Figure 16 plants-14-01911-f016:**
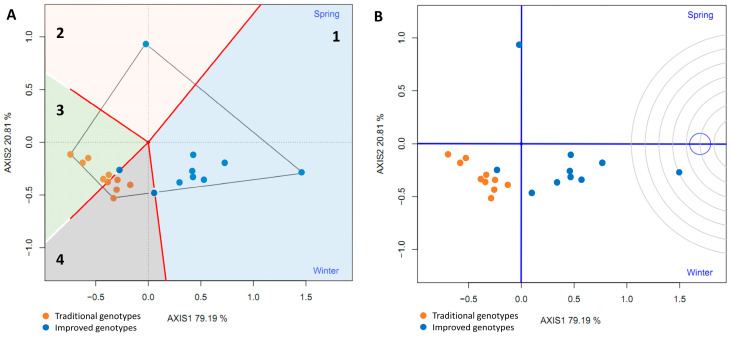
GGE biplot “Which Won Where/Whats” (**A**) and environment ranking (**B**) based on the comparison of the performance of 20 sweet potato varieties (traditional and improved) and two environments (spring and winter) (4 replicates).

**Figure 17 plants-14-01911-f017:**
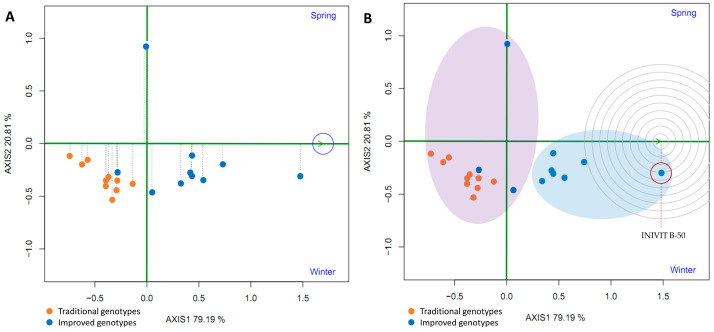
GGE biplot average yield and stability of genotypes (**A**) and ranking of genotypes (**B**) based on the comparison of the yield of 20 sweet potato varieties (traditional and improved) and two environments (spring and winter) (4 replicates).

**Figure 18 plants-14-01911-f018:**
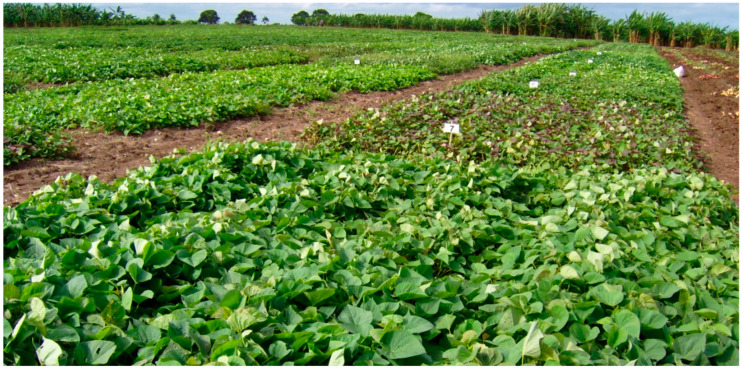
Completely randomized block design with five replicates.

**Table 1 plants-14-01911-t001:** Combined analysis of variance for some sweet potato cultivars traits in two seasons of the year.

Source of Variance	df	Sum Sq	Mean Squares	F-Value	*p*-Value
Yield					
Season	1	1419	1419	30.338	<0.001 ***
Cultivars	1	8788	8788	187.924	<0.001 ***
Season x Cultivars	1	19	19	0.417	0.52
Residuals	116	5424	47		
Commercial yield					
Season	1	948	948	23.93	<0.001 ***
Cultivars	1	6153	6153	155.38	<0.001 ***
Season x Cultivars	1	39	39	0.98	0.32
Residuals	116	4594	40		
Number of tuberous roots					
Season	1	2.02	2.02	0.589	0.444
Cultivars	1	92.93	92.93	27.099	<0.001 ***
Season x Cultivars	1	0	0	0.001	0.97
Residuals	116	397.8	3.43		
Foliage yield					
Season	1	594.1	594.1	26.583	<0.001 ***
Cultivars	1	184.6	184.6	8.261	<0.01 **
Season x Cultivars	1	7.4	7.4	0.33	0.56
Residuals	116	2592.6	22.3		
Dry matter					
Season	1	91.3	91.3	14.685	<0.001 ***
Cultivars	1	17.7	17.67	2.842	0.094
Season x Cultivars	1	0	0.02	0.003	0.95
Residuals	116	721.2	6.22		
°Brix					
Season	1	11.2	11.224	3.889	0.051
Cultivars	1	5.6	5.59	1.937	0.167
Season x Cultivars	1	0.5	0.547	0.189	0.664
Residuals	116	334.8	2.886		

Significance codes: *** *p* < 0.001; ** *p* < 0.01.

**Table 2 plants-14-01911-t002:** Traditional and improved sweet potato cultivars.

Traditional Cultivars		Improved Cultivars	
Cultivar	Year of Release	Cultivar	Year of Release
‘Maní’	Unknown *	‘CEMSA 74-228’	1974
Baracutey’	1969	‘CEMSA 78-354’	1981
‘Cocullo Blanco’	Unknown	‘CEMSA 78-326’	1981
‘Cuba-1’	1969	‘INIVIT B 98-2’	1998
‘Haití’	Unknown	‘INIVIT B 98-3’	1998
‘Duclos XI’	1969	‘INIVIT B2-2005’	2005
‘EVA’	Unknown	‘INIVIT BS-16’	2006
‘Campanilla’	Unknown	‘INIVIT B-16’	2010
‘Boca de saco’	Unknown	‘INIVIT B-23’	2013
‘Blanco Redondo’	1966	‘INIVIT B-50’	2015

* Traditional cultivars lack release records due to historical data gaps.

**Table 3 plants-14-01911-t003:** Sweet potato varieties evaluated by period.

Year of Release	Cultivar	Year of Release	Cultivar
1970 (before)	‘Baracutey’	1991–2000	‘INIVIT B 98-2’
	‘Cuba-1’		‘INIVIT B 98-3’
	‘Duclos XI’		‘INIVIT B 98-4’
	‘Blanco Redondo’		‘INIVIT B 98-7’
1971–1980	‘CEMSA 74-228’	2001–2010	‘INIVIT B2-2005’
	‘CEMSA 72-1’		‘INIVIT BS-16’
	‘CEMSA 74-195’		‘INIVIT B-240’
	‘CEMSA 74-298’		‘INIVIT B-61’
1981–1990	‘CEMSA 78-354’	2011–2020	‘INIVIT B-23’
	‘CEMSA 78-326’		‘INIVIT B-65’
	‘CEMSA 78-425’		‘INIVIT B-50’
	‘CEMSA 80-77’		‘INIVIT BM-90’

**Table 4 plants-14-01911-t004:** Monthly accumulated rainfall and average monthly relative humidity at INIVIT from November 2023 to March 2024.

Climate Variables	November	December	January	February	March
Precipitation (mm)	72.3	69.0	52.1	53.3	42.0
Relative humidity (%)	83	82	80	76	74

## Data Availability

The raw data supporting the conclusions of this article will be made available by the authors on request.

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
