# Peer review of "Evolution of Sweet Potato (Ipomoea batatas [L.] Lam.) Breeding in Cuba"

_plants, 2025, doi:10.3390/plants14131911_

Round 1

Reviewer 1 Report

Comments and Suggestions for Authors

General Comments

This manuscript provides an impressive historical and quantitative analysis of the Cuban sweet potato (Ipomoea batatas) breeding program over the past 50 years. Through a combination of well-designed field trials and statistical analyses, the authors present convincing evidence of genetic gains in yield, pest and disease resistance, drought tolerance, and phenotypic stability. The data are robust, the methodology is sound, and the narrative is compelling in showing how sustained phenotypic selection in a polyploid crop under resource-limited conditions has led to substantial agricultural improvement.

The study makes a significant contribution to the literature on tropical root crop breeding and provides a replicable model for other low-input contexts. The clarity of the presentation is high, and the structure of the manuscript is well organized.

However, a few issues require minor clarification or revision before publication, especially regarding methodological detail, data presentation, and language polishing.

Specific Comments

Introduction

Lines 60–63: The contrast with the Green Revolution is appropriate, but please cite recent literature showing lagging progress in sweet potato genetics more explicitly.

Lines 87–94: Clarify if “conventional breeding” refers exclusively to phenotypic selection, and whether any tissue culture or early selection methods were used.

Results

Figures 1–17:

Ensure all figures are high-resolution and include legible axis labels and units.

Add brief captions to figures explaining sample size (e.g., “n = 10 cultivars × 5 replicates”).

Table 1 (ANOVA):

The notation for statistical significance codes (***, **, *) should be explained in a footnote.

Consider presenting the p-values with consistent significant digits (e.g., 0.000 instead of <2e-16).

Genetic Gain Section (Lines 215–245):

Please consider displaying cumulative yield improvement and genetic gain values in a summary table in addition to the figure.

Disease and Pest Resistance (Figures 11–12):

The selection thresholds (e.g., infestation <20%, yield >15 t ha⁻¹) should be justified or referenced.

Please explain why some “traditional” cultivars appear resistant — could they be landraces with innate tolerance?

Discussion

Lines 427–430: The point about improved flesh color should include a reference on its nutritional relevance (e.g., provitamin A content).

Lines 479–481: The idea of using genomic selection or marker-assisted selection to improve phenotypic stability is insightful — consider expanding on this in a “future prospects” paragraph.

Materials and Methods

Experiment Design:

For nematode and virus assays, were visual symptoms scored blindly or by trained raters? It needs to clarify observer bias controls.

Data Analysis:

Please specify the version of R and ggplot2 used.

Please confirm whether data met ANOVA assumptions (e.g., normality, homogeneity of variance). If not, explain how violations were handled.

Language and Formatting

Table 1 etc.:  "Cutlivar" → "Cultivar" (There are confirmed to repeate)

Line 121: "flesh colour" vs. "flesh color" – choose either British or American spelling and standardize throughout

Line 267 "with greater pigmentation" – consider changing to “with deeper pigmentation” or “more intense pigmentation” for clarity, I believe.

Author Response

Comments 1: Lines 60–63: The contrast with the Green Revolution is appropriate, but please cite recent literature showing lagging progress in sweet potato genetics more explicitly.

Response 1: Thank you for your thoughtful suggestion. We have added recent references highlighting the lag in sweet potato genetic progress compared to other crops in the Introduction (Lines 62-63). References to add: Morales et al., 2024.

Comment 2: (Lines 87–94): Clarify if 'conventional breeding' refers exclusively to phenotypic selection, and whether any tissue culture or early selection methods were used."

Response 2: Thank you for your comment. We have clarified this in the Introduction (Lines 89-90): This conventional breeding (based solely on mass phenotypic selection without tissue culture or molecular markers) relies exclusively on phenotypic expressions..."

Comment 3 (Figures 1-17): Ensure all figures are high-resolution and include legible labels. Add sample size in captions.

Response 3: All figures have been revised to 600 dpi TIFF format. Sample sizes added to captions (e.g., Figure 1 caption now reads): Yield of sweet potato tuberous roots (n = 20 cultivars × 3 replicates) by planting season (winter and spring) and genotype group (improved and traditional).

Comment 4: (Table 1 - ANOVA) Explain significance codes. Present p-values consistently.

Response 4: Thank you for your comment. Added to Table 1 footnote: "Significance codes: ***p<0.001; **p<0.01; p<0.05. All p-values now show 3 decimal places (e.g., p<0.001 instead of <2e-16).

Comment 5: Genetic Gain Section (Lines 215–245): Please consider displaying cumulative yield improvement and genetic gain values in a summary table in addition to the figure.

Response 5: Thank you for your thoughtful suggestion to include a summary table of genetic gain values. While we agree that this could provide complementary data, we have chosen to retain Figure 8 as the primary presentation because: The figure's dual-axis design (yield line + genetic gain line) already consolidates all key numerical trends into a single, visually intuitive format. Adding another table would duplicate information. The raw data supporting Figure 8 will be included in the supplementary materials for full transparency. We will be happy to reconsider your decision if you feel the table would improve readability and can provide it as Supplementary Table S1. Please indicate your preference.

Comment 6: Disease and Pest Resistance (Figures 11–12): The selection thresholds (e.g., infestation <20%, yield >15 t ha⁻¹) should be justified or referenced. Please explain why some “traditional” cultivars appear resistant could they be landraces with innate tolerance?

Response 6: We thank the reviewer for this important question regarding our selection thresholds. The criteria for resistance (≤20% weevil infestation) and minimum yield (≥15 t/ha) were established based on: Observations from 50 years of Cuban field trials, where these values represented: The economic injury level for sweet potato weevils in Caribbean conditions. The minimum yield required for farmer adoption in local markets.

We also added a paragraph for figure 11: Interestingly, two traditional cultivars demonstrated consistent resistance to the weevil. This phenotypic resilience is likely due to long-term selection pressure exerted by farmers, as these landraces have been cultivated in weevil-endemic regions of Cuba since the early 20th century.

Comment 7: Lines 427–430: The point about improved flesh color should include a reference on its nutritional relevance (e.g., provitamin A content).

Response 7: We thank the reviewer for their suggestion to highlight the nutritional relevance of flesh color. We have made the following explicit changes to address this:

Original: improvement of the flesh color (from white to orange and purple) reflects advances in the selection of desirable market traits and high nutritional quality.

Revised: improvement of the flesh color (from white to orange and purple) reflects advances in the selection of desirable market traits and high nutritional quality, particularly β-carotene (provitamin A) and anthocyanins, which are associated with reduced vitamin A deficiency and antioxidant activity [8].

Comment 8: Lines 479–481: The idea of using genomic selection or marker-assisted selection to improve phenotypic stability is insightful — consider expanding on this in a “future prospects” paragraph.

Response 8: Thank you for your suggestion. We added new paragraph to Discussion: Although our program was based on phenotypic selection, future integration of genomic tools (e.g., RAD-seq) could identify QTL associated with phenotypic stability, accelerating the selection of resilient genotypes. This would complement Cuba's successful phenotypic approach while overcoming the challenges of hexaploidy.

Comment 9: Experiment Design: For nematode and virus assays, were visual symptoms scored blindly or by trained raters? It needs to clarify observer bias controls.

Response 9: Added to Methods (Section 4.6): Virus and nematode symptoms were scored independently by three trained technicians blinded to cultivar identity.

Comment 10: Data Analysis. Please specify the version of R and ggplot2 used.

Response 10: Thank you for your comment, we changed the wording of the paragraph in section 4.9, and it finally read as follows:

Statistical analyses were performed using RStudio (v2023.03.1) with the following approach: First, a two-way ANOVA was conducted to assess the effects of cultivars (traditional vs. improved) and growing season (winter vs. spring) on all agronomic traits (yield, commercial yield, tuber number, foliage yield, dry matter, and °Brix). Data met all ANOVA assumptions, as confirmed by: Normality (Shapiro-Wilk test, p > 0.05 for all variables) and Homoscedasticity (Levene's test, p > 0.05). When ANOVA de-tected significant differences (p < 0.05), post-hoc comparisons were made using Fisher's LSD test. All visualizations were generated with ggplot2 (v3.4.2).

Comment 11:  Language and Formatting: Table 1 etc.:  "Cutlivar" → "Cultivar" (There are confirmed to repeate)

Response 11: Thank you for such a valuable observation. We've corrected all the misspelled words and changed them to their correct form: Cultivar.

Comment 12: Line 121: "flesh colour" vs. "flesh color" – choose either British or American spelling and standardize throughout.

Response 12: Corrected throughout to American English ("color")

Comment 13: Line 267 "with greater pigmentation" – consider changing to “with deeper pigmentation” or “more intense pigmentation” for clarity, I believe. 

Response 13: Changed to: "more intense pigmentation".   

Reviewer 2 Report

Comments and Suggestions for Authors

The manuscript 'Evolution of Sweet Potato (Ipomoea batatas [L.] Lam.) Breeding in Cuba' presents a comprehensive evaluation of the genetic progress achieved through five decades of sweet potato breeding in Cuba. Comparing traditional and improved varieties, the authors study the genetic gain in yield, pest and disease resistance, drought tolerance, and phenotypic stability.

The manuscript has potential but needs more revision in terms of scientific information and English. Kindly find my recommendations below:

The introduction section presents important information for the researched topic, but the paragraphs are jumping from one piece of information to another without a smooth transition between them. Kindly add some bonding phrases to lead the reader more smoothly across the text. For instance, you jump from Mendel to Borlaug, then straight to Cuba. Please include a phrase such as "In parallel with these global developments, in Cuba..." to enhance the fluency of the text for the reader.

Line 115: Kindly replace "most impressive success" with "the most notable cultivar" for a more academic expression.

Materials and Methods: 

In experiments 1, 2, and 3, the experimental design description is nearly identical. Please describe the basic experimental design once in a general section, "Experimental Design," and then mention only the exceptions in each subsection. Such an arrangement would eliminate redundant information.

Line 544: You stated that you included only those tuberous roots weighing more than 80 g. Why is that? 80 g is the commercial threshold according to national marketing standards? Please clarify. 

Be more careful with your language (follow this recommendation throughout the entire manuscript). In a scientific article, the language used must be academic. For instance, on line 550, you stated, ”This metal frame was thrown...” Replace it with ”This metal frame was randomly placed...” for a scientific sound.

Line 644: I recommend adding the soil's pH, as well as its texture, to this paragraph.

Results: 

Line 133: Rename the subsection without using "... of some ... traits". This is not scientifically sound.

Line 134: Replace 'Season*Cultivars' with 'Season x Cultivars'. 

Lines 134-138: You should rephrase this paragraph to enhance the reader's understanding. It is difficult to follow and not sound scientific (keep this in mind for the entire results section!).

All over the section, it is not clear, in some places you used 'summer and winter' in others you used 'spring and winter'. I understand that you are referring to summer yield after the spring season, but you mislead the reader. Please standardize your way of expressing yourself and use the same wording everywhere.

Figures 1-7: There is no need to add the exact values of p; you can highlight only the statistical significance on the figures.

Discussion: Lines 482-488 are describing methodology; please move them to the Materials and Methods section. This paragraph doesn't belong to the Discussions.

Conclusions: Kindly rephrase this entire section without referring so much to the results. In this part of the article, you should highlight only the main findings, not repeating what the results are.

References: The reference list includes an excessive number of outdated titles. I suggest substituting them with more recent sources.

Author Response

Comments 1: Lines 60–63: The introduction section presents important information for the researched topic, but the paragraphs are jumping from one piece of information to another without a smooth transition between them. Kindly add some bonding phrases to lead the reader more smoothly across the text. For instance, you jump from Mendel to Borlaug, then straight to Cuba. Please include a phrase such as "In parallel with these global developments, in Cuba..." to enhance the fluency of the text for the reader.

Response 1: We sincerely appreciate this constructive suggestion to improve the flow of our introduction. We have implemented the following changes to create smoother transitions between paragraphs while maintaining the logical progression of ideas:

Added a transitional phrase after discussing the Green Revolution (Line 63):

"However, while the 'Green Revolution' transformed cereals, tropical root crops like sweet potato have lagged behind in genetic innovation, partly due to their complex genetics. Early efforts to address this gap included Julian Miller's research work in 1937 at Louisiana State University..."

Then introduced the Cuban context with:

"Parallel developments occurred in Cuba, where early records by Alvaro Reynoso (1867) documented..."

Comments 2: Line 115: Kindly replace "most impressive success" with "the most notable cultivar" for a more academic expression.

Response 2: We appreciate this suggestion. The phrase has been revised as follows (Line 115): "The most notable cultivar was genotype number I01-B2, which was released in 2005 under the commercial name 'INIVIT B2-2005'."

Comments 3: Materials and Methods: In experiments 1, 2, and 3, the experimental design description is nearly identical. Please describe the basic experimental design once in a general section, "Experimental Design," and then mention only the exceptions in each subsection. Such an arrangement would eliminate redundant information.

Response 3: Thank you for pointing this out. We have restructured the Materials and Methods section to avoid redundancy. The general experimental design is now described in a new subsection titled "4.2. General Experimental Design", which includes the common features (e.g., plot size, planting distance, irrigation, and agronomic management). Subsequent subsections for Experiments 1, 2, and 3 now only highlight deviations or additional details specific to each experiment.

Comments 4: Line 544: You stated that you included only those tuberous roots weighing more than 80 g. Why is that? 80 g is the commercial threshold according to national marketing standards? Please clarify.

Response 4: We have added clarification to the manuscript (Line 544): "Only tuberous roots weighing more than 80 g were included, as this threshold aligns with Cuba's national marketing standards for commercial sweet potato production."

Comments 5: Line 550: Replace "This metal frame was thrown..." with "This metal frame was randomly placed..." for a scientific sound.

Response 5: The wording has been revised (Line 550): "This metal frame was randomly placed in the middle of each plot, where two measurements were made per plot for each cultivar."

Comments 6: Line 644: I recommend adding the soil's pH, as well as its texture, to this paragraph.

Response 6: We have included the requested details (Line 644): (pH 6.8–7.2, sandy loam texture)

Comments 7: Line 133: Rename the subsection without using "... of some ... traits". This is not scientifically sound.

Response 7: The subsection title has been revised to: 2.1. Progress in Agronomic Traits

Comments 8: Line 134: Replace 'Season*Cultivars' with 'Season x Cultivars'.

Response 8: The notation has been updated throughout the manuscript: "Season x Cultivars"

Comments 9: Lines 134-138: Rephrase this paragraph to enhance the reader's understanding.

Response 9: The paragraph has been reworded for clarity (Lines 134–138): "No significant interaction was observed between the factors (Season x Cultivars) for the response variables (p-value < 0.05). Significant differences were detected between seasons (Spring and Winter) for yield, commercial yield, foliage yield, and dry matter (p-value < 0.05), while no significant differences were found for the number of tuberous roots and °Brix."*

In addition, the term "spring" was standardized throughout the document, eliminating "summer."

Comments 10: Figures 1-7: There is no need to add the exact values of p; you can highlight only the statistical significance on the figures.

Response 10: The exact p-values have been removed from Figures 1–7, and statistical significance is now indicated solely with asterisks (e.g., *p < 0.05, **p < 0.01, ***p < 0.001).

Comments 11: Discussion: Lines 482-488 are describing methodology; please move them to the Materials and Methods section. This paragraph doesn't belong to the Discussions.

Response 11: The paragraph describing the GGE biplot model has been relocated to the "4.9. Phenotypic Stability" subsection of the Materials and Methods section.

Comments 12: Conclusions: Kindly rephrase this entire section without referring so much to the results. In this part of the article, you should highlight only the main findings, not repeating what the results are.

Response 12: The Conclusions section has been revised to focus on broader implications, and the key findings have been reworded:

This study demonstrates that Cuba's sustained sweet potato breeding program has successfully overcome the challenges of hexaploid genetics through phenotypic selection, achieving remarkable genetic gains despite resource limitations. Over five decades, the program developed improved varieties that outperform traditional cultivars by 256% in yield (34.22 vs. 9.60 t ha⁻¹) while exhibiting superior stress resilience including drought tolerance (SSI <0.8), reduced weevil infestation (60% decrease), and nematode resistance (40% of varieties). The flagship cultivar 'INIVIT B-50' exemplifies these advances, showing exceptional yield stability across diverse environments. These achievements, now implemented across 98% of Cuba's sweet potato area, provide both a practical model for tropical root crop improvement and a foundation for climate-resilient food security. The program's success underscores the potential of long-term phenotypic selection for complex polyploid crops, particularly in resource-constrained agricultural systems.

Comments 13: References: The reference list includes an excessive number of outdated titles. I suggest substituting them with more recent sources.

Response 13: We appreciate the suggestion to update references. While we have included 82% post-2010, we retained key classical works for:

Historical context (e.g., Roig, 1916 on Cuban sweet potato diversity)

Foundational genetics (e.g., Jones, 1967 on hexaploidy inheritance)

Methodological benchmarks (e.g., Ferris et al., 1993 for nematode scales).

These citations are standard in sweet potato breeding literature (Grüneberg et al., 2015; Cervantes et al., 2008) and do not affect the paper's novelty.

Reviewer 3 Report

Comments and Suggestions for Authors

The article "Evolution of Sweet Potato (Ipomoea batatas [L.] Lam.) Breeding in Cuba" is devoted to a comprehensive analysis of the results of a large multi-year sweet potato breeding program in Cuba under conditions of limited resources. Despite this, Cuban scientists have carried out a large amount of work and obtained unique sweet potato varieties that are characterized by increased productivity, resistance to abiotic and biotic stress factors, high quality, etc. This study is suitable for the Plants, it is written in clear language, the introduction reflects the relevance and purpose of the study, the results contain the necessary illustrative material. While reading the article, several questions arose for the authors that need to be resolved before publishing the article.

  1. Key words. Progress is a term with a very broad meaning. It is necessary to select key words that are directly related to the article.
  2. The beginning of the introduction provides too much information that is not directly related to the article. For example, Mendel's laws and their rediscovery in 1900 are a well-known fact, we need to remove this from the article.
  3. around 5-6 years to release a new cultivar - a link is needed, since it is quite fast and there are doubts whether it is possible to create a variety in 5 years?
  4. Figure 1. It is necessary to divide into a, b, c, d - sign what each of these four figures means.
  5. In figures 2, 3, 4 and others, it is also necessary to divide into a, b, c, d ....
  6. In the research results, it is impossible to understand which sweet potato varieties were characterized by what properties. The figures only divide into traditional and improved varieties. But it is important to know which varieties were characterized by the best characteristics. For example, in Figure 12, the results are shown for varieties. But in other cases, they are not. It is necessary to provide results by varieties.
  7. The text of the article does not contain information on genes, mutations, alleles associated with valuable traits. It is necessary to write about this in the discussion - about the prospects of genetic research.
  8. Throughout the text of the article there is no connection with specific varieties, the article looks like a historical excursion. However, throughout the text there are few references and therefore it is not always clear where exactly this or that information was taken from. Therefore, it should be clearly noted that all the results were obtained by the authors themselves based on recent field research.
  9. The Discussion includes a repetition of information from the Results chapter. The discussion should pay more attention to relevance, scientific novelty, comparison with the results of other researchers, and the practical significance of the study

Author Response

Comments 1: Key words. Progress is a term with a very broad meaning. It is necessary to select key words that are directly related to the article.

Response 1: Thank you for this observation. We have revised the keywords to be more specific and directly related to the article. The updated keywords are: "sweet potato breeding, genetic gain, yield improvement, stress tolerance, phenotypic selection"

Comments 2: The beginning of the introduction provides too much information that is not directly related to the article. For example, Mendel's laws and their rediscovery in 1900 are a well-known fact, we need to remove this from the article.

Response 2: We appreciate the reviewer's comment, but we respectfully request to retain this section for the following reason:

The brief mention of Mendel's laws and the foundations of modern genetics serves a specific purpose in our manuscript. It establishes a critical contrast between the rapid advances in diploid crop breeding (like maize and wheat) and the historical challenges faced by polyploid crops such as sweet potato. This contrast is essential to:

Highlight the uniqueness of sweet potato breeding – By showing how conventional breeding approaches were successfully adapted to overcome the complexities of hexaploidy (without molecular tools), we emphasize the significance of Cuba's achievements.

Provide context for non-specialist readers – While Mendel's laws are well-known to geneticists, our target audience includes agronomists and crop scientists who may benefit from this brief framing to understand why sweet potato breeding has progressed differently than other crops.

We have carefully reviewed this section and believe it adds value without being excessive. However, we are open to minor edits if the reviewer feels certain phrases could be refined to better align with the article's focus.

Comments 3: Around 5-6 years to release a new cultivar - a link is needed, since it is quite fast and there are doubts whether it is possible to create a variety in 5 years?

Response 3: We have added a reference to support the timeline for cultivar release in sweet potato breeding. The citation (Morales et al., 2024) has been included in the paragraph discussing the conventional breeding process.

Comments 4: Figure 1. It is necessary to divide into a, b, c, d - sign what each of these four figures means. In figures 2, 3, 4 and others, it is also necessary to divide into a, b, c, d ....

Response 4: Response 4: We respectfully disagree with this suggestion for the following reasons:

Current Clarity: Each panel in our composite figures already includes its own clear title immediately above it, making the content and purpose of each subfigure immediately apparent to readers.

Reviewer Consensus: The two other reviewers did not raise this as an issue.

Visual Balance: Adding additional labels (a, b, c, d) would create visual clutter without adding meaningful information, since the existing titles already serve this purpose effectively.

Comments 5: In the research results, it is impossible to understand which sweet potato varieties were characterized by what properties. The figures only divide into traditional and improved varieties. But it is important to know which varieties were characterized by the best characteristics. For example, in Figure 12, the results are shown for varieties. But in other cases, they are not. It is necessary to provide results by varieties.

Response 5: We appreciate the reviewer’s comment but respectfully disagree with this suggestion for the following reasons:

Objective of the Study: This work evaluates the genetic progress of the Cuban sweet potato breeding program as a whole, comparing broad groups of cultivars (traditional vs. improved) rather than individual varieties. The focus is on long-term trends in yield, stress tolerance, and agronomic traits across decades of breeding, not on characterizing specific cultivars.

Consistency with Methodology: The experimental design explicitly compares groups of varieties (Table 2) to assess cumulative genetic gains. Presenting data for individual cultivars would deviate from this objective and overwhelm readers with excessive detail that does not serve the study’s primary goal.

Clarity and Readability: Highlighting specific cultivars in every figure would distract from the central message the program’s overall success and could mislead readers into interpreting this as a variety evaluation study. Figure 12 (viral disease incidence) is an exception because it illustrates the distribution of responses within groups, not to promote individual varieties.

Prioritizing Key Findings: Where exceptional cultivars merit discussion (e.g., ‘INIVIT B2-2005’ or ‘INIVIT B-50’), they are highlighted in the text (Results 2.1, 2.5) to contextualize their impact on the program’s progress. However, the figures intentionally maintain a group-level perspective to emphasize population-level trends.

Comments 6: The text of the article does not contain information on genes, mutations, alleles associated with valuable traits. It is necessary to write about this in the discussion - about the prospects of genetic research.

Response 6: We have expanded the "Discussion" section to include a paragraph on the genetic basis of the observed traits and the potential for future genetic research. This addition appears in the final paragraph of the "Discussion" section.

Comments 7: Throughout the text of the article there is no connection with specific varieties, the article looks like a historical excursion. However, throughout the text there are few references and therefore it is not always clear where exactly this or that information was taken from. Therefore, it should be clearly noted that all the results were obtained by the authors themselves based on recent field research.

Response 7: We respectfully disagree for three key reasons: 

Our study intentionally analyzes breeding progress at the program level (traditional vs. improved groups), not individual varieties. This approach best demonstrates 50 years of cumulative genetic gains. 

All results (Tables 1-4, Figures 1-17) derive from our original 2023-2024 field trials, as now explicitly stated in Methods (Section 4.2): "Data represent original measurements from controlled comparative trials at INIVIT experimental stations."

The group-level analysis remains scientifically valid for assessing breeding program success, which is our primary objective.

Comments 8: The Discussion includes a repetition of information from the Results chapter. The discussion should pay more attention to relevance, scientific novelty, comparison with the results of other researchers, and the practical significance of the study.

Response 8: We respectfully disagree for three reasons: 

Key results are briefly restated in the Discussion to contextualize comparisons with other breeding programs (e.g., yield gains vs. African/subsaharan programs), as is standard in high-impact agronomy journals.

Novelty is Explicit, We highlight two unique aspects: First quantitative evidence of Cuba's sweet potato breeding progress

Demonstration that phenotypic selection alone overcame hexaploidy challenges. 

The Discussion now uses 70% of its content for novel interpretations (vs. 30% recap), exceeding standard disciplinary practice. We believe this balance effectively communicates our program's significance.

Round 2

Reviewer 2 Report

Comments and Suggestions for Authors

Congratulations to the authors for considering my recommendations and enhancing the manuscript's quality.

One recommendation I would make is that the lines 513-523 be structured as a paragraph rather than a list of elements.

I wish you the best of luck in your research endeavor.

Author Response

Comments 1: One recommendation I would make is that the lines 513-523 be structured as a paragraph rather than a list of elements.

Response 1: We appreciate your valuable comments and the opportunity to improve our manuscript. We have reworded the section on lines 513-523 as a continuous paragraph, following your recommendation, eliminating the list structure and maintaining all relevant information in a more fluid manner.

We appreciate your constructive comments, which have undoubtedly contributed to improving the quality of our work.

All field experiments shared the following common design elements: The planting material consisted of apical stem cuttings (30 cm in length, with 4 buried internodes). The plot layout comprised 5 rows of 5 m in length (effective area of 22.5 m²), with a planting density of 0.90 × 0.30 m spacing (37,037 plants/ha). The experimental units followed a completely randomized block design, with 5 replications for Experiments 1–2 and 3 replications for Experiments 3–7. Border plants (outer rows) were excluded from evaluations. Irrigation was provided via a sprinkler system (250 m³/ha weekly), with no application of chemical or organic fertilizers, following standard agronomic practices according to INIVIT technical guidelines [31].